# A Partition Based Gradient Compression Algorithm for Distributed Training in AIoT

**DOI:** 10.3390/s21061943

**Published:** 2021-03-10

**Authors:** Bingjun Guo, Yazhi Liu, Chunyang Zhang

**Affiliations:** Department of Computer Science and Technology, North China University of Science and Technology, Tangshan 063210, China; youngwind1995@gmail.com (B.G.); zchunyang571@gmail.com (C.Z.)

**Keywords:** AIoT, distributed training, gradient compression, training efficiency

## Abstract

Running Deep Neural Networks (DNNs) in distributed Internet of Things (IoT) nodes is a promising scheme to enhance the performance of IoT systems. However, due to the limited computing and communication resources of the IoT nodes, the communication efficiency of the distributed DNN training strategy is a problem demanding a prompt solution. In this paper, an adaptive compression strategy based on gradient partition is proposed to solve the problem of high communication overhead between nodes during the distributed training procedure. Firstly, a neural network is trained to predict the gradient distribution of its parameters. According to the distribution characteristics of the gradient, the gradient is divided into the key region and the sparse region. At the same time, combined with the information entropy of gradient distribution, a reasonable threshold is selected to filter the gradient value in the partition, and only the gradient value greater than the threshold is transmitted and updated, to reduce the traffic and improve the distributed training efficiency. The strategy uses gradient sparsity to achieve the maximum compression ratio of 37.1 times, which improves the training efficiency to a certain extent.

## 1. Introduction

With the rapid development of the Internet of Things (IoT), it has become a trend to apply AI technology to the IoT, and creates a new concept, AIoT(Artificial Intelligence & Internet of Things). AIoT refers to the application of artificial intelligence (AI) algorithms in the Internet of Things (IoT) to achieve a higher level of the intelligent system of the Internet of Things. In an AIoT system, it is usually composed of many sensors and cloud servers to collect, store and process data. At the same time, with the rapid development of the information age, higher requirements are put forward for the data processing and low delay communication of equipment in the AIoT system. However, due to a large number of sensors, small size, and limited computing power, the ability to access and communicate with each other is limited, and it is necessary to extend their service life and minimize the number of maintenance [1,2]. Therefore, how to improve the communication efficiency between the sensor nodes and improve the overall serviceability has become one of the urgent problems to be solved.

In this paper, we model the above problem as a distributed training problem in deep learning. As shown in Figure 1, the left half of Figure 1 describes the AIoT architecture, where multiple sensors are connected to Cloud Computing through the network layer. In the right half of the figure, we map AIoT to a distributed training model. Each sensor in AIoT is equivalent to a worker node in a distributed training scene, which is responsible for calculating the gradient and transmitting it to the server; and Cloud Computing in AIoT is equivalent to a server node in a distributed training scene, responsible for summarizing global gradients and updating model parameters. Due to the relatively weak communication capabilities of IoT devices, the gradient exchange efficiency between various nodes has become one of the important bottlenecks that restrict the quality of AIoT. To solve this problem, this paper proposes a gradient partition-based compression algorithm, Partition Gradient Compression (PGC). PGC can improve the overall training efficiency by reducing the communication cost between distributed nodes and improving communication efficiency, thereby improving the efficiency of distributed training in AIoT.

In recent years, deep learning technology has attracted much attention due to its outstanding performance in many artificial intelligence fields such as image recognition, natural language processing, and speech recognition. With the increasing scale of deep learning and the increasing complexity of neural network structure, distributed deep learning becomes more and more important in research and practice. The communication efficiency of gradient exchange is a major obstacle to the improvement of distributed training efficiency [3], therefore, how to improve the communication efficiency of distributed training is also one of the hot spots in current research.

One of the most direct ways to deal with communication costs is to reduce traffic, or gradient compression [4]. Solutions based on gradient compression can be divided into two categories: gradient quantization and gradient sparsity. Gradient quantization reduces communication costs by using fewer bits to reduce the number of gradients transmitted. However, the compression ratio based on the quantization method is limited and can only be compressed to 1-bit at most.

The sparse method is to select a threshold value and send gradient elements larger than the threshold value, while gradient values smaller than the threshold value are not sent. Although this method can obtain a high compression ratio, it often damages the accuracy of the model. For example, Han et al. [5] proposed the Deep Gradient Compression method, in which the Gradient was sparse to 0.1% of the original gradient and the Compression rate could reach a maximum of 600 times. However, due to the high proportion of sparsity, four strategies such as momentum correction, local gradient pruning, momentum factor masking, and warm-up training have to be adopted to compensate for the loss of accuracy. Although this method achieves a high compression ratio, the correction cost is also very high, but the training task quantity is increased.

In this paper, we propose an adaptive compression method based on gradient partition to overcome the limitations of the above methods. We analyzed the distribution characteristics of the gradient in each layer of the neural network and concluded that the distribution of the gradient was regular and approximately normal. Similar conclusions were also found in references [6,7]. According to the gradient distribution characteristics, the gradient region is divided, to screen out more favorable gradient values for training, to realize the reasonable distribution of updated gradient values in different gradient division intervals. Based on gradient partition, we further introduce a method of adaptive threshold selection for gradient update. Different from the existing method of setting threshold by experience, this method combines the information entropy of gradient distribution to select a reasonable threshold to realize threshold filtering. It is also proved that the gradient descent method can achieve normal convergence based on gradient partition. It can be said that on the premise of ensuring the accuracy and convergence of training, our method improves the efficiency of training to a certain extent.

## 2. Related Works

At present, researchers have proposed many methods to reduce communication costs and speed up distributed training of neural networks. Among them, gradient compression is one of such methods. Compared with other solutions, gradient compression strategy only plays a role in the communication content between computing nodes but does not affect the communication process, let alone change the structure of the training model. Therefore, to a certain extent, gradient compression achieves the purpose of reducing communication costs. Solutions based on gradient compression strategy can be roughly divided into two categories: gradient quantization and gradient sparsity.

### 2.1. Gradient Quantization

Gradient quantization refers to the quantization of gradient and the reduction of its precision. The principle of quantization is to reduce the transmission of data by reducing the bit of the gradient value of floating-point type transmitted, to improve the communication efficiency between nodes. At present, the main challenge of gradient quantization is how to obtain satisfactory model accuracy with less gradient information. Researchers have proposed several methods for quantifying gradients.

Wen et al. [8] proposed the Ternary Gradients(TernGrad) algorithm, which quantified the gradient value into three values $\left\{-1,0,1\right\}$ to reduce the cost of gradient communication. It is also proved that the convergence of the algorithm can be guaranteed as long as the quantized gradient mean is equal to the original gradient. Through TernGrad algorithm, the gradient of the traffic can be reduced to 1/20 of the original.

Khirirat et al. [9] studied quantization under deterministic setting conditions, that is, gradient values can be calculated noiselessly, and proved that parallel gradient descent with unbiased quantization will converge to the optimal solution of a strongly convex function. Seide et al. [10] proposed a 1-bit quantization method to quantify each gradient parameter as a symbol bit, that is, in the communication between computing nodes, each parameter is quantized as a bit, which can significantly reduce the size of communication data. At the same time, to avoid the decline of the accuracy of the model, the error in the quantization process is retained and added to the calculation gradient in the next step. This method achieves a 10-fold acceleration ratio in the classical speech recognition application.

In order to solve the problem of low model accuracy or low compression rate in the existing quantization schemes, Guo et al. [11] proposed a novel adaptive quantization scheme(AdaQS). AdaQS algorithm according to the ratio of the mean and standard deviation of the gradient (MSDR) automatically determine the quantitative level, and achieve a balance between model accuracy and quantitative level.

Mishchenko et al. [12] considered the stochastic setting in data parallelization and showed that Stochastic Gradient Descent (SGD) with unbiased quantization converges to near the stationary points of non-convex and strongly convex functions. However, the Quantized SGD (QSGD) method proposed by Alistarh et al. [13] is to achieve a balance between training accuracy and gradient quantization by rounding the gradient into a linear quantization value randomly, to maintain a certain training accuracy without quantizing the result of excessive damage to the training. At the same time, the validity and convergence of the method can be guaranteed in the process of gradient quantization.

### 2.2. Gradient Sparsification

Gradient sparsity reduces communication costs by setting a threshold and sending a small number of parameters. Aji et al. [14] proposed a distributed gradient descent method, in which the gradient is sparse by removing the gradient with the minimum absolute value of R% (R% is a fixed percentage). At the same time, to ensure convergence, a layer with a global threshold is used for normalization. This method can map 99% of the minimum gradient value to zero and improve the speed of translation by 11% when applied to neural machine translation without affecting the translation quality.

Shi et al. [15] discussed that under the influence of communication bandwidth in distributed training, both the gradient sparse method and the gradient merge method would be damaged to a certain extent, and therefore could not achieve a better training effect. They proposed an SGD-based gradient sparse optimization algorithm (OMGS-SGD). The algorithm considers the tradeoff between communication and computation to find the optimal solution. The experimental results show that the efficiency of the proposed algorithm is improved by 31% when the convergence is almost the same as that of the original SGD.

The Adaptive Residual Gradient Compression (AdaComp) method proposed by Chen et al. [16] divides the gradient residuals of each layer into several groups of fixed size, and use the gradient with the largest absolute value in each group as the threshold to screen the gradient, thus using local gradient activities to automatically adjust the compression rate. If the previous gradient residual combined with the gradient value calculated in proportion exceeds this maximum value, the gradient is considered important and the gradient is updated, otherwise, it is not updated.

Strom et al. [17] developed a threshold sparse method, in which only the gradient greater than the set threshold is sent, thus only a small part of the value of the transmission gradient is needed to be updated. Similar to 1-bit quantization, the residual gradient after sparsity is accumulated into the next gradient update. At the same time, how to determine the threshold value of the gradient is also a thorny problem. Dryden et al. [18] proposed a fixed proportion of positive and negative gradients to send gradients instead of fixed thresholds.

In this section, two methods to reduce communication costs were introduced, gradient quantization and gradient sparsity. Gradient quantization is realized by reducing the bit of gradient transmission, but the compression rate is limited. Gradient sparsity improves the compression rate by setting a threshold to screen the gradient, but it is a difficult problem how to select a reasonable threshold. In this paper, we propose an adaptive compression method based on gradient partition, which divides the gradient according to its distribution characteristics and selects an appropriate threshold value for gradient update in combination with the information entropy of gradient distribution. At the same time, our method will not affect the gradient communication process, let alone change the training model structure.

## 3. Gradient Distribution

To solve the problems mentioned in the previous section, we started from the distribution characteristics of the gradient, and after investigating relevant literature, we reached the conclusion that the gradient distribution in the neural network is approximately normal distribution. This can be verified in references [6,7]. In references [6], Wang et al. set up experiments and trained on AlexNet network by using ImageNet, CIFAR10, and other data sets, respectively, concluded conclusion that the gradient was an approximately normal distribution. Similarly, Chmiel et al. In references [7] also achieved similar results by using the dataset ImageNet on ResNet18 network. These papers all proved the conclusion that the gradient distribution in neural network training is an approximately normal distribution.

Based on the above references, the characteristics of gradient distribution are re-counted and analyzed in this paper. In this paper, the deep neural network VGG16 and CIFAR10 datasets were used for training. Besides, the log files generated by training are visualized through the TensorBoard tool provided in TensorFlow, a deep learning framework, and the gradient distribution results are obtained (see the results in Figure 2).

In Figure 2, the gradient distribution of some layers of VGG16 is shown.Due to too many layers, only the gradient distribution characteristic map of some layers of VGG16 is selected. Where the upper left corner of each graph indicates the gradient distribution of which layer in VGG16. Meanwhile, the horizontal axis represents the values of the gradients and the vertical axis represents the statistical quantity of the gradient value in this layer.

To sum up, both the conclusions of relevant literature and the conclusions of experiments in this paper can fully explain that the gradient distribution in the neural network is approximately normal distribution. According to the property of normal distribution, it can be known that about 68.26% of the samples fall between (μ−σ,μ+σ). This means that most of the gradient values are distributed around 0 (when μ = 0), and this part of gradient parameters has little influence on the training. Therefore, the problem of redundancy is generated. Therefore, gradient sparsity is necessary.

## 4. Compression Based on Gradient Partition

According to the gradient distribution characteristics obtained in the previous section, the gradient distribution is approximately described as *x* ~ *N*(μ,$\sigma^{2}$). According to the gradient of the distribution of the following division (see the division in Figure 3).

According to the characteristics of gradient update, choosing a gradient value with a larger absolute value is more beneficial to update and has a greater influence on the weight. Therefore, we take the (μ−3σ,μ−σ)∪(μ+σ,μ+3σ) interval of the gradient distribution as the focus area of the gradient update and the interval of (μ−σ,μ+σ) is the sparse area of the gradient update. The method that can be considered is to take more gradient values in the key interval of the gradient update to participate in the update, while in the sparse interval of the gradient update, tries to take fewer values (or even no values in special cases) to participate in the update.

Based on gradient division, this paper uses information entropy to measure the importance of gradient information in each interval, so as to serve as the basis for gradient selection.

### 4.1. Information Entropy of Normal Distribution

Information entropy is often used to measure the uncertainty of variable distribution. The following is the information entropy calculation formula under the probability distribution of discrete variables [19]:(1)$$H\left(X\right)=-\sum_{i=1}^{n}p\left(x_{i}\right)logp\left(x_{i}\right)$$

The probability density function of normal distribution is:(2)$$p\left(x\right)=\frac{1}{\sqrt{2\pi }\sigma }exp\left(-\frac{{\left(x-\mu \right)}^{2}}{2\sigma^{2}}\right)$$

Then the Shannon information of normal distribution is (the base of logarithm is *e*):(3)$$-lnp\left(x\right)=\frac{1}{2}\left(ln\left(2\pi \right)+2ln\left(\sigma \right)+{\left(\frac{x-\mu }{\sigma }\right)}^{2}\right)$$

The information entropy of normal distribution, that is, the expectation of Shannon information. Therefore, the information entropy of normal distribution is:(4)$$E=\frac{1}{2}\left(ln\left(2\pi \right)+2ln\sigma +1\right)$$

Considering that the gradient values in the neural network are all decimals, to ensure that the information entropy is positive, the expression is expressed as follows:(5)$$H=-\frac{1}{2}\left(ln\left(2\pi \right)+2ln\sigma +1\right)$$

It can be seen from the above formula that the information entropy of normal distribution is only related to its standard deviation.

### 4.2. Threshold Setting

In this section, we propose a gradient threshold screening algorithm based on information entropy to set the threshold screening gradient. The specific process of the algorithm is shown in Algorithm 1.
**Algorithm 1** Sparse gradient based on information entropy algorithm.**Require:** 
  Gradients $G_{l}$, *l* = 1 … *L* for *l*-th layer in the net;**Require:** 
  The number of gradient elements in the layer *N*;**Ensure:** 
  $T_{l}$, *l* = 1 … *L*;1:**for** layer *l* = 1 … *L*
**do**2:  Calculate the average of $G_{l}$:$\bar{g_{l}}=\frac{\sum_{l=1}^{N}g_{l}}{N};$3:  Calculate the standard deviation of $G_{l}$:$\sigma_{l}=\sqrt{\frac{1}{N}\sum_{l=1}^{N}{\left(g_{l}-\bar{g_{l}}\right)}^{2}};$4:  Calculate the information entropy of the normal distribution: $H=-\frac{1}{2}\left(ln\left(2\pi \right)+2ln\sigma_{l}+1\right);$5:  Function: $H\left(\alpha \right)=-\frac{1}{2}\left(ln\left(2\pi \right)+2ln\sigma_{\alpha }+1\right);$6:  
$\gamma =abs[\left(G_{max}^{l}-G_{min}^{l}\right)/G_{sum}^{l}];$
7:  
S=H(α)∗γ;
8:  Threshold: $T_{l}=Top-k\;\left(G_{l},S\right);$9:**end for**

The algorithm has the following important steps. Firstly, what we need to do is how to select the appropriate gradient threshold according to the information entropy *H* of gradient distribution. The specific description is as follows, let the gradient threshold be α, and the constructor: (6)$$H\left(\alpha \right)=-\frac{1}{2}\left(ln\left(2\pi \right)+2ln\sigma_{\alpha }+1\right)$$
where $\sigma_{\alpha }$ is the standard deviation of the gradient in the layer where the threshold is located, and *H(α)* is the information entropy of the gradient distribution in the layer.
(7)$$\gamma =abs[\left(G_{max}^{l}-G_{min}^{l}\right)/G_{sum}^{l}]$$

Secondly, where γ is the trade-off factor, $G_{max}^{l}$ is the maximum gradient value of each layer, $G_{min}^{l}$ is the minimum gradient value in each layer, and $G_{sum}^{l}$ is the sum of absolute gradient values of each layer.
(8)S=H(α)∗γ

Thirdly, where *S* is the set gradient sparsity. According to the gradient sparsity, we use the *Top-k* sorting algorithm to find the first *S* large gradient values in this layer, and get the threshold α.$G_{l}$ is the gradient value of *l*-th layer.
(9)$$Threshold=Top-k(G_{l},S)$$

Finally, the function $T_{\alpha }\left(g_{l}\right)$ represents the threshold selection process, where $g_{l}$ represents the gradient elements in *l*-th layer and α is the set threshold value. Then the gradient screening process can be represented by the following piecewise functions:(10)$$T_{\alpha }\left(g_{l}\right)=\left\{\begin{array}{cc} g_{l} & |g_{l}|>\alpha \\ \alpha & |g_{l}|=\alpha \\ 0 & |g_{l}|<\alpha \end{array}\right.$$

It is through this way to filter and update the gradients in training.

### 4.3. Convergence Analysis of Gradient Partition

In this section, because of the division of the gradient distribution interval, the convergence analysis is used to prove whether the gradient descent in the partition interval can achieve convergence without affecting the training. We get the following conclusion:(11)$$L\left(w_{k}\right)\leq L\left(w_{*}\right)+\frac{\eta }{2}\alpha^{2}$$

Preparation condition. Given the training data set of size *n*, any one random sample of the training data set is represented as φ(x,y), the model parameter *w* is defined, and the loss function is represented as *l(w,φ)*, then the objective function is minimized as:(12)$$L\left(w\right)=\frac{1}{n}\sum_{i=1}^{n}l\left(w,\varphi_{i}\right)$$

If $\varphi_{k}$ is the sample input for *k* iterations, then the random gradient value is expressed as follows, the random gradient value can be abbreviated as $g(w_{k})$.
(13)$$g\left(w,\varphi_{k}\right)=\frac{1}{n_{k}}\sum_{i=1}^{n_{k}}▽l\left(w,\varphi_{k_{i}}\right)$$

**Theorem 1.** 
*Gradient function ▽L(w) is on the interval (μ−3σ,−|α|)∪(|α|,μ+3σ), and the Lipschitz condition is satisfied for the nonconvex object L(w) and the Lipschitz constant λ > 0. Lipschitz continuity condition, is a more smooth condition than usual, which is often used to analyze the convergence of gradient descent algorithm. Let the step size be η, the initial value $w_{0}$, and the number of iterations k ∈ N (N is the set of natural numbers), $w_{k+1}=w_{k}+\eta ▽L(w)$. If L(w) converges, then the following equation is satisfied:*


(14)$$L\left(w_{k}\right)\leq L\left(w_{*}\right)+\frac{\left|\right|w_{0}-w_{*}{\left|\right|}^{2}}{2\eta_{t}k}$$

Let $w_{*}=argmaxL\left(w\right)$ be the optimal solution, if the Lipschitz condition is satisfied for the nonconvex target *L(w)*, then for the step $\eta_{t}$:(15)$$\eta_{t}\leq \frac{1}{\lambda }$$

The λ is the Lipschitz constant, then (14) can be written as follows:(16)$$L\left(w_{k}\right)\leq L\left(w_{*}\right)+\frac{\lambda ||w_{0}-w_{*}{\left|\right|}^{2}}{2k}$$

**Proof.** Suppose that the gradient function ▽*L(w)*: $R\to R^{m}$ is Lipschitz continuous and the Lipschitz constant λ > 0, for any $w_{k},w_{k+1}\in R^{m}$:
(17)$$L\left(w_{k+1}\right)\leq L\left(w_{k}\right)+▽L\left(w_{k}\right)\left(w_{k+1}-w_{k}\right)+\frac{\lambda }{2}\left|\right|w_{k+1}-w_{k}{\left|\right|}^{2}$$According to the gradient descent method $w_{k+1}=w_{k}-\eta ▽L\left(w_{k}\right)$, the right side of Formula (17) can be written as:
$$\begin{array}{cc} \; & =L\left(w_{k}\right)+▽L\left(w_{k}\right)\left(-1\right)\left(\eta ▽L\left(w_{k}\right)\right)+\frac{\lambda }{2}\eta^{2}▽L^{2}\left(w_{k}\right)\\ \; & =L\left(w_{k}\right)-\eta ▽L^{2}\left(w_{k}\right)+\frac{\lambda }{2}\eta^{2}▽L^{2}\left(w_{k}\right)\\ \; & =L\left(w_{k}\right)-\eta \left(1-\frac{\lambda }{2}\right)▽L^{2}\left(w_{k}\right) \end{array}$$
$$\begin{array}{cc} \; & \leq L\left(w_{k}\right)-\frac{\eta }{2}▽L^{2}\left(w_{k}\right)\left(\eta \leq \frac{1}{\lambda }\right)\\ \; & \leq L\left(w_{*}\right)+▽L\left(w_{k}\right)\left(w_{k}-w_{*}\right)-\frac{\eta }{2}\left|\right|▽L\left(w_{k}\right){\left|\right|}^{2}\\ \; & \left(usedL\left(w_{k}\right)-L\left(w_{*}\right)\leq ▽L\left(w_{k}\right)\left(w_{k}-w_{*}\right)\right)\\ \; & =L\left(w_{*}\right)+\frac{w_{k}-w_{k+1}}{\eta }\left(w_{k}-w_{*}\right)-\frac{\eta }{2}\left|\right|\frac{w_{k}-w_{k+1}}{\eta }{\left|\right|}^{2}\\ \; & \left(usedw_{k+1}=w_{k}-\eta ▽L\left(w_{k}\right)\right) \end{array}$$By expanding and merging the above formula, we can get the following results:
(18)$$\begin{array}{cc} & =L\left(w_{*}\right)+\frac{1}{2\eta }{\left(w_{k}-w_{*}\right)}^{2}-\frac{1}{2\eta }\left(|\right|w_{k}-w_{*}{\left|\right|}^{2}-2\eta ▽L\left(w_{k}\right)\left(w_{k}-w_{*}\right)+||\eta ▽L\left(w_{k}\right){\left|\right|}^{2})\\ & =L\left(w_{*}\right)+\frac{1}{2\eta }{\left(w_{k}-w_{*}\right)}^{2}-\frac{1}{2\eta }\left(|\right|w_{k}-w_{*}\left|\right|-||\eta ▽L\left(w_{k}\right){\left||\right)}^{2} \end{array}$$According to $w_{k}-w_{k+1}=\eta ▽L\left(w_{k}\right),▽L\left(w_{k}\right),▽L\left(w_{k+1}\right)\in \left(\mu -3\sigma ,-|\alpha |\right)\cup \left(|\alpha |,\mu +3\sigma \right)$, therefore $w_{k}-w_{k+1}\in \eta \left(\mu -3\sigma ,-|\alpha |\right)\cup \eta \left(|\alpha |,\mu +3\sigma \right)$.
(19)$$\begin{array}{cc} \; & =L\left(w_{*}\right)+\frac{1}{2\eta }{\left(\eta |\alpha |\right)}^{2}-\frac{1}{2\eta }\left(||\eta |\alpha ||\right|-{\left||\eta |\alpha |||\right)}^{2} \end{array}$$
(20)$$\begin{array}{cc} \; & =L\left(w_{*}\right)+\frac{\eta }{2}\alpha^{2} \end{array}$$In summary:
(21)$$L\left(w_{k+1}\right)\leq L\left(w_{*}\right)+\frac{\eta }{2}\alpha^{2}$$
$$L\left(w_{k+1}\right)-L\left(w_{*}\right)\leq \frac{\eta }{2}\alpha^{2}$$Each iteration is expanded by formula:
$$L\left(w_{1}\right)-L\left(w_{*}\right)\leq \frac{\eta }{2}\alpha^{2}$$
$$L\left(w_{2}\right)-L\left(w_{*}\right)\leq \frac{\eta }{2}\alpha^{2}\;\ldots \;\ldots$$
$$L\left(w_{k}\right)-L\left(w_{*}\right)\leq \frac{\eta }{2}\alpha^{2}$$
Add up the above formula:
$$\sum_{k=1}^{k}L\left(w_{k}\right)-kL\left(w_{*}\right)\leq \frac{k\eta }{2}\alpha^{2}$$
$$L\left(w_{k}\right)-L\left(w_{*}\right)\leq \frac{\eta }{2}\alpha^{2}$$
(22)$$L\left(w_{k}\right)\leq L\left(w_{*}\right)+\frac{\eta }{2}\alpha^{2}$$
So far, we get a conclusion (22). At the same time, the convergence condition is as follows:
$$L\left(w_{k}\right)\leq L\left(w_{*}\right)+\frac{\left|\right|w_{0}-w_{*}{\left|\right|}^{2}}{2\eta_{t}k}$$If $w_{k}-w_{k+1}\in \eta \left(\mu -3\sigma ,-|\alpha |\right)\cup \eta \left(|\alpha |,\mu +3\sigma \right)$. They are as follows:
$$L\left(w_{k}\right)\leq L\left(w_{*}\right)+\frac{{\left||\eta |\alpha ||\right|}^{2}}{2\eta_{t}k}$$
(23)$$L\left(w_{k}\right)\leq L\left(w_{*}\right)+\frac{\eta }{2k}\alpha^{2}$$□

$\frac{\eta }{2}\alpha^{2}$ in Formula (22) > $\frac{\eta }{2k}\alpha^{2}$ in Formula (23), $\frac{\eta }{2}\alpha^{2}$,$\frac{\eta }{2k}\alpha^{2}$ are both positive numbers, the following formula must be established.
$$L\left(w_{k}\right)\leq L\left(w_{*}\right)+\frac{\left|\right|w_{0}-w_{*}{\left|\right|}^{2}}{2\eta_{t}k}$$

Therefore, the gradient function must converge on the interval (μ−3σ,−|α|)∪(|α|,μ+3σ).

## 5. Experiment

In order to verify the effectiveness of the adaptive compression algorithm based on gradient partition proposed in this paper, an experimental simulation environment was constructed to simulate the distributed training scene of neural networks for experimental verification. Through simulation experiments we verified the performance of our gradient compression algorithm, from the convergence of the algorithm, compression ratio, and training throughput to analyze and evaluate.

The datasets used in this experiment were MNIST and Cifar10, and the parameter information of the neural network models used in training is shown in Table 1.

The experiment in this paper was completed by TensorFlow framework [20], which was implemented in the form of single machine multi-card. The Parameter Server (PS) node and Worker nodes in distributed training were simulated by three port addresses of the local machine. One port address was used to simulate the PS node, which was mainly responsible for summarizing and calculating the global gradient and updating the model parameters. The other two port addresses were used to simulate the Worker node to calculate the gradient and receive the latest model parameters from the PS node. The bandwidth of links in the network was set to 10 Mbit/s. At the same time, the data set used in this paper is MNIST, and the neural network models used included LeNet, AlexNet and ResNet.

### 5.1. Convergence Analysis

The convergence of the algorithm refers to the algorithm after many iterations, the numerical value tends to a certain value, the convergence of the algorithm reflects the ability of the algorithm to find the optimal solution. The accuracy rate of the three algorithms under the AlexNet network model is shown in Figure 4, where Baseline represents the reference group, Gradient Variance represents the compression algorithm based on gradient variance [21], and Partition Gradient Compression (PGC) represents the adaptive compression algorithm based on gradient partition proposed in this paper.

It can be seen from Figure 4 that the algorithm proposed in this paper had a similar accuracy to the Baseline, Gradient Variance, and other methods. With the increase of the number of training steps, the overall fluctuation of the curve waas more stable, which indicates that the threshold sparsity played a certain role. Although there was some fluctuation in the early iteration, it was since in the initial stage of training, the threshold value of gradient was directly sparse, which would cause certain accuracy damage. In the loss curve, the loss was smaller than the other two methods.

The three algorithms were also trained in the ResNet network model. The accuracy and loss curves in ResNet network model are shown in Figure 5.

ResNet network adopted the ResNet-50 and MNIST datasets. Under the same experimental conditions, the accuracy of model recognition could reach more than 95% after only 300 steps of training. As can be seen from Figure 5, compared with the other two methods, the overall fluctuation of the adaptive compression algorithm based on gradient partition was more stable, and the accuracy rate gradually tended to a stable range, indicating that the algorithm could achieve normal convergence, and the loss curve also showed that the algorithm could achieve normal convergence.

Similarly, training on the LeNet network could obtain similar results, as shown in Figure 6.

According to the curve fluctuation in Figure 6, it can be seen that the adaptive compression algorithm based on gradient partition could also achieve normal convergence. Besides, we also verified the convergence of AlexNet and ResNet on the dataset Cifar10, as shown in Figure 7 and Figure 8.

It can be seen from Figure 7 and Figure 8 that the PGC method in this paper also achieved better accuracy and convergence effect on different datasets. To sum up, by analyzing the convergence of the algorithm on the three network models, the algorithm proposed in this paper could achieve good convergence under the same experimental conditions. At the same time, compared with the other two methods, it also achieved better contrast. In this section, the training accuracy of all network models are shown in Table 2.

### 5.2. Compression Ratio Analysis

In this paper, the data size of the total gradient parameters in the training of three kinds of network models was counted, and the gradient data size after compression by related algorithms was counted as the measurement of algorithm compression ratio. The details are shown in Table 3.

It can be seen from the above table that the compression ratio of the algorithm increased with the increase of network layers by testing on three different network models. In the AlexNet model, the algorithm proposed in this paper could achieve a compression ratio of about 29.1 times, while for the network model ResNet with more layers, it could achieve a compression ratio of about 37.1 times. When training AlexNet and ResNet on Cifar10,the compression ratio could reach 25.6 times and 32.9 times. Compared with the other two methods, the compression ratio was also improved.

### 5.3. Training Throughput Analysis

According to the analysis of training throughput of adaptive compression algorithm based on gradient partition, the training throughput refers to the number of batch data read in unit time. The larger the batch value, the faster the reading speed and the higher the training efficiency. This experiment was carried out under the condition of two computing nodes and the bandwidth was 10 Mbit/s. The training throughput of the proposed algorithm on three different network models was tested. The training throughput results are shown in Figure 9.

Through the analysis of the above histogram, it can be seen that although the efficiency of the algorithm was limited in the LeNet network with few network layers, it may also have been affected by the bandwidth of communication and the size of computing tasks. However, compared with the other two methods, the training throughput was improved to a certain extent. In addition, the throughput of our method on the AlexNet network was increased by 1.18 times compared with the Gradient Variance method and 1.4 times higher than that of the Baseline method. When training on the network model ResNet with more layers, the efficiency of the algorithm was improved by 1.3 times compared with Gradient Variance method and 1.5 times higher than Baseline method.

## 6. Conclusions

In this paper, we analyze and study the distribution characteristics of gradient parameters in neural network training for AIoT systems, and propose an adaptive compression strategy based on gradient partition. Firstly, according to the distribution characteristics of the gradient, the gradient value is divided into the key area of gradient update and the sparse area. At the same time, the information entropy formula of gradient distribution feature is used to select the gradient value. To reduce the communication cost and improve the training efficiency, the gradient is sparse to a certain extent by selecting the updated gradient value in the partition by the threshold.

Through the analysis of relevant experimental results, our method can achieve normal convergence after the training of different network models, and compared with the method without compression scheme, this algorithm can improve the compression ratio to a certain extent, and under the premise of ensuring the accuracy and convergence, the maximum compression ratio can achieve about 37.1 times. Besides, the algorithm also improves the training throughput and speeds up the training speed to a certain extent. At the same time, the method proposed in this paper also provides some more useful ideas for data transmission between nodes in Mobile Edge Computing (MEC) and communication optimization in Federated Learning (FL). We also hope that this idea can be applied to more fields and provide more meaningful progress.

## Figures and Tables

**Figure 1 sensors-21-01943-f001:**
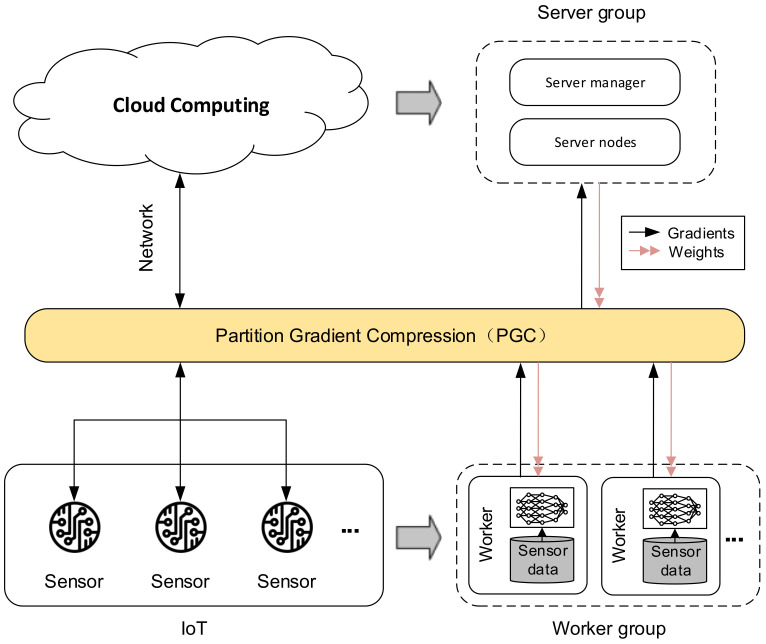
The training scene in IoT is modeled as distributed training.

**Figure 2 sensors-21-01943-f002:**
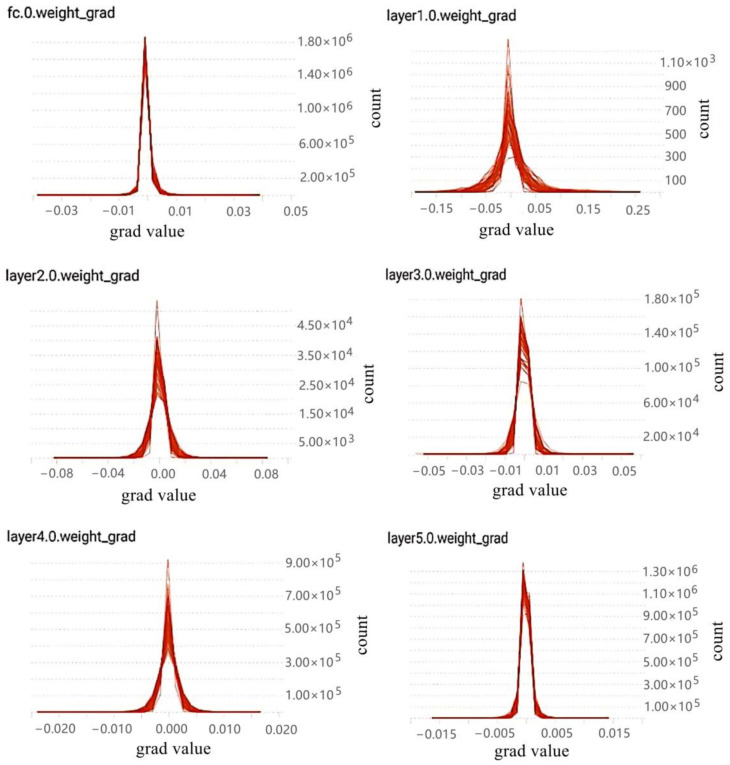
Gradient distribution histogram of some layers sampled in VGG16.

**Figure 3 sensors-21-01943-f003:**
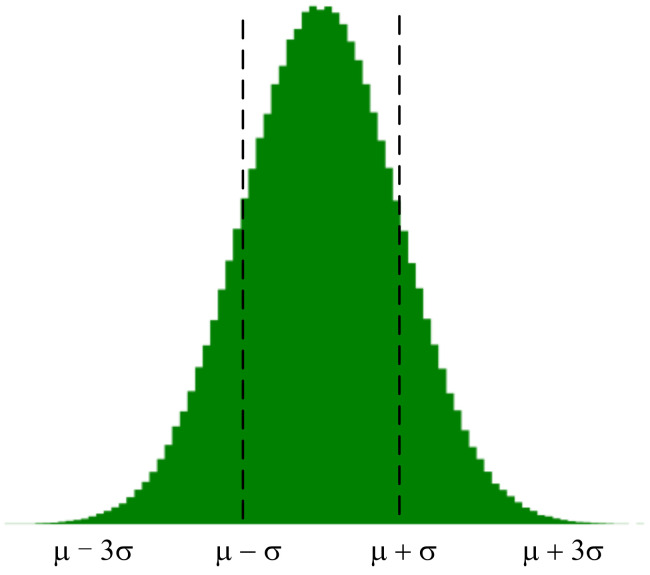
Gradient partition.

**Figure 4 sensors-21-01943-f004:**
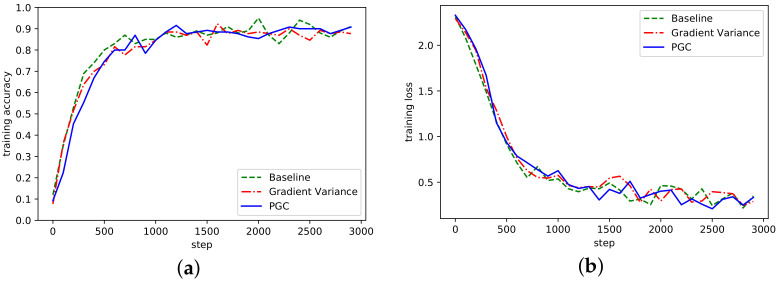
(**a**) Accuracy in AlexNet network on MNIST; (**b**) Loss curve in AlexNet network on MNIST.

**Figure 5 sensors-21-01943-f005:**
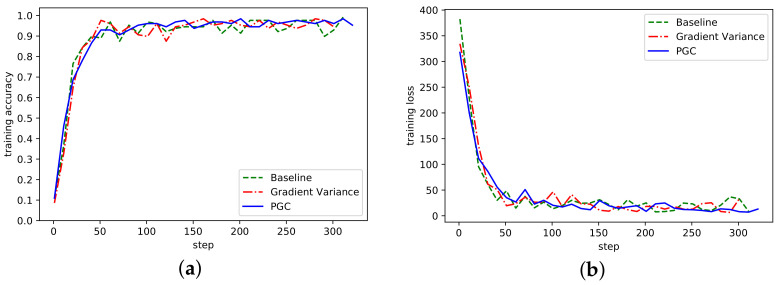
(**a**) Accuracy in ResNet network on MNIST; (**b**) Loss curve in ResNet network on MNIST.

**Figure 6 sensors-21-01943-f006:**
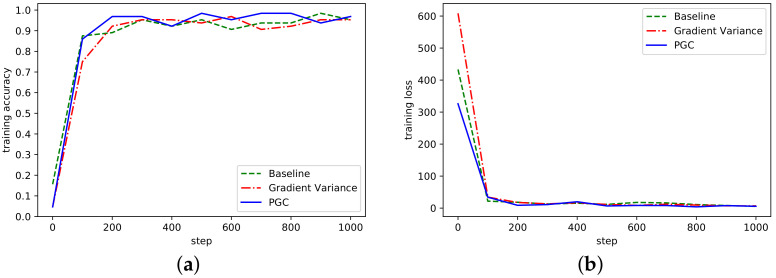
(**a**) Accuracy in LeNet network on MNIST; (**b**) Loss curve in LeNet network on MNIST.

**Figure 7 sensors-21-01943-f007:**
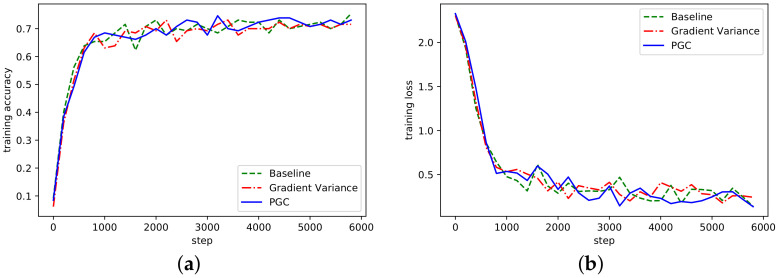
(**a**) Accuracy in AlexNet network on Cifar10; (**b**) Loss curve in AlexNet network on Cifar10.

**Figure 8 sensors-21-01943-f008:**
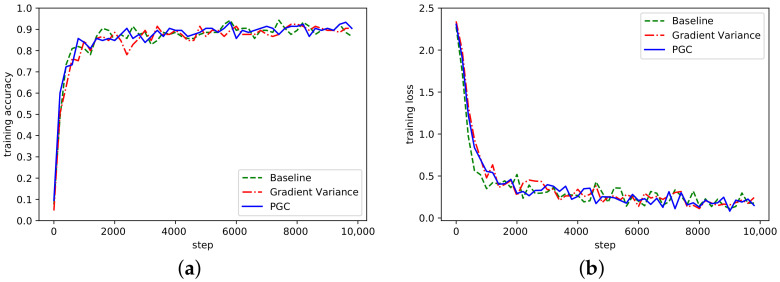
(**a**) Accuracy in ResNet network on Cifar10; (**b**) Loss curve in ResNet network on Cifar10.

**Figure 9 sensors-21-01943-f009:**
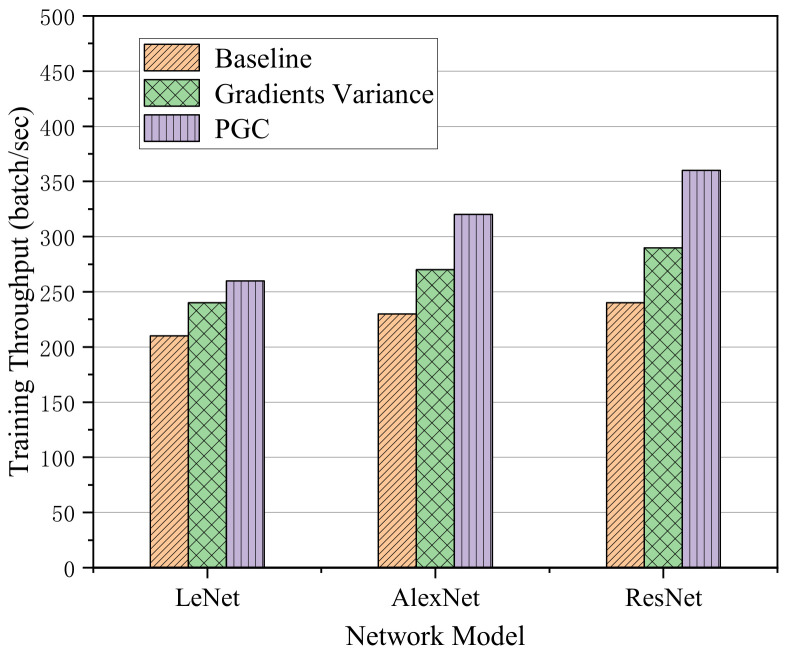
Training throughput on three different network models.

**Table 1 sensors-21-01943-t001:** Parameter information in training.

Dataset	Network Model	Batch Size	Iterations	Learning Rate
	LeNet	128	1000	0.01
MNIST	AlexNet	128	3000	0.01
	ResNet	128	300	0.01
Cifar10	AlexNet	128	6000	0.001
	ResNet	128	10,000	0.001

**Table 2 sensors-21-01943-t002:** The accuracy of all network models.

Datasets	Network Model	Baseline	Gradient Variance	PGC
	LeNet	96.65%	96.59%	96.86%
MNIST	AlexNet	90.02%	89.95%	91.17%
	ResNet	95.90%	96.45%	96.75%
Cifar10	AlexNet	71.78%	71.89%	72.21%
	ResNet	89.97%	90.25%	90.43%

**Table 3 sensors-21-01943-t003:** Compression ratio information.

Dataset	Network Model	Gradient Size	Baseline	Gradient Variance	PGC
	LeNet	0.45 M	1×	18.3×	24.2×
MNIST	AlexNet	1.39 M	1×	23.7×	29.1×
	ResNet	4.27 M	1×	30.4×	37.1×
Cifar10	AlexNet	4.56 M	1×	20.1×	27.3×
	ResNet	6.50 M	1×	25.6×	32.9×

## Data Availability

Not applicable.

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
