# Peer review of "A Partition Based Gradient Compression Algorithm for Distributed Training in AIoT"

_sensors, 2021, doi:10.3390/s21061943_

Round 1

Reviewer 1 Report

First of all my comments are turned to the application of the proposed algorithm. It is stated in the article (with the help of related work) that it addresses IoT with low resources. Just looking on the amount of communicated date and bandwidth someone would notice that the main applicability would remain distributed computers without power consumption constrains and with high bandwidth communications.  

Some comments:

Row 99 – the first sentence lacks in meanings. Please rephrase it. How is the quantization scheme low, or who is low? Which article proposes the AdaQS, Guo et al or yours?! It is not clear form the text.

Row 105 – SGD acronym should be “unfolded”! (i.e. stochastic gradient descent)

Row 138 – I believe that “are introduced” should be changed to “were introduced” as they were introduced above in that specific section.

Fig. 1 is not enough explained – the gradient is multidimensional. Even VGG16 is considered a well-known CNN model, some details regarding how those distribution were represented are mandatory. The axis deserver proper description the same as the distributions themselves as there are more distributions on the same plot. “the number of statistics” of which gradient? (row 165)

In ec. (1) there is no problem as omitting the base of the logarithm, but not the same thing applies for equation (3) which stands only for natural logarithm (ln).

Ec. (6) – from a discrete variable it is switched to a continues variable which needs proper discussion.

Row 193 – The chi square distribution is introduced without the proper context.

The entire 4.1. section should be either minimized to the result by referring to other work which contain a correct proof or rewritten with a correct demonstration.

Algorithm 1 – there are missing variables or parameters involved. 1. Who is N? Who is g i in comparison with G (I understand that g I is the components of G but there are more Gs and this is not clearly specified as required when an algorithm is described)? What line 5 represent in an algorithm, or line 7 if line 5 would be a function?! Line 8, who is  function k( )?

Row 220 who is phi(x,y)?! random sample of whom?

The entire mathematical approach needs a revision according to the requirements of such scientific communication. I skipped the 4.3 paragraph until the paper will be revised in accordance with necessary rigor.

In conclusion, the article discusses an important topic (distributed training), it provides one proposed approach (algorithm) which should be described clear, it provides a testing environment which is hard to reproduce and somehow not edifying for the intended  purpose of the algorithm (only two nodes which could not create bottleneck on communication, the synchronization is simple and it is intrinsic to the Tensorflow, thus, it is not quite what it would be needed).  The other algorithms against which the PGC was compared are not listed to show the “implementation” according to which the results where obtained. In order for the article to have value in this hot topic the experiments should be described as it can be reconstructed by other researcher the same as this article was able to implement the algorithms of other researchers.  

Author Response

Point 1: Row 99 – the first sentence lacks in meanings. Please rephrase it. How is the quantization scheme low, or who is low? Which article proposes the AdaQS, Guo et al or yours?! It is not clear form the text.

Response 1: Thank you very much for your suggestion. We corrected the related description. Reference [12] pointed out the problems of low model accuracy and low compression rate in the current quantization scheme, so they proposed an AdaQS scheme to solve the problem. In order to avoid confusion with the author's name, we complete the name of the author to distinguish our paper.

Point 2: Row 105 – SGD acronym should be “unfolded”! (i.e. stochastic gradient descent)

Response 2: Thank you very much for your correction. We have launched SGD for Stochastic Gradient Descent.

Point 3: Row 138 – I believe that “are introduced” should be changed to “were introduced” as they were introduced above in that specific section.

Response 3: Thank you very much for your correction. We have changed "are introduced" to "were introduced".

Point 4: Fig. 1 is not enough explained – the gradient is multidimensional. Even VGG16 is considered a well-known CNN model, some details regarding how those distribution were represented are mandatory. The axis deserver proper description the same as the distributions themselves as there are more distributions on the same plot. “the number of statistics” of which gradient? (row 165)

Response 4: We are sorry to confuse you, and thank you for your questions. Fig.1 is to visualize the network layer in VGG16 with the help of TensorFlow's TensorBoard tool. Fig.1 shows the gradient distribution of the weights in these layers. As for which layer, the upper left corner of each subgraph in Fig. 1 is marked with that layer, for example, layer1.0.weight_grad represents the gradient distribution of the weight value of layer 1.0 in VGG16.

Of course, there is also layer1.1.weight_grad, etc. This mark form comes with TensorBoard visualization tool. It visualizes all layers of VGG16 with tags like this. Since VGG16 has too many layers, we only take the gradient distribution of the weights of some layers to form Fig. 1.

Meanwhile, the horizontal axis represents the values of the gradients, and the vertical axis represents the statistical quantity of the gradient value in this layer. At the same time, we also add the meaning of abscissa and ordinate to the picture for better understanding. See Figure 2 after modification.

Point 5: In ec. (1) there is no problem as omitting the base of the logarithm, but not the same thing applies for equation (3) which stands only for natural logarithm (ln).

Response 5: Thank you very much for your correction. We have found this problem and restated that the base of logarithm of this formula is e, and ln is used to replace the previous logarithm log.

Point 6: Ec. (6) – from a discrete variable it is switched to a continues variable which needs proper discussion.

Response 6: We are sorry to confuse you, and thank you for your suggestion. We split Ec. (3) into Ec. (4) and (5) just for the convenience of calculating their expectations. Ec. (4) and (5) are originally part of Ec. (3), so Ec. (6) does not involve the conversion from discrete variables to continuous variables, they exist together.

Point 7: Row 193 – The chi square distribution is introduced without the proper context.

Response 7: Thank you very much for your suggestion. We have supplemented the introduction of chi square distribution with degree of freedom of 1.

Point 8: The entire 4.1. section should be either minimized to the result by referring to other work which contain a correct proof or rewritten with a correct demonstration.

Response 8: Thank you very much for your suggestion. We have corrected 4.1 section and look forward to meeting your suggestions.

Point 9: Algorithm 1 – there are missing variables or parameters involved. 1. Who is N? Who is g i in comparison with G (I understand that g I is the components of G but there are more Gs and this is not clearly specified as required when an algorithm is described)? What line 5 represent in an algorithm, or line 7 if line 5 would be a function?! Line 8, who is  function k( )?

Response 9: We are very sorry for the ignored problems, and thank you for your questions. We have made some supplements, N represents the number of gradient elements in l-th layer,

 represents the gradient elements in l-th layer , but there is a mistake here, it should be . In line 5 represents a function, in line 7 not a function. To avoid confusion, we have changed S (a) to S, besides, because there is a problem with the form of writing, k() should actually be Top-k().

Point 10: Row 220 who is phi(x,y)?! random sample of whom?

Response 10: We are so sorry to confuse you, and thank you for your question. We complete the meaning of , it represents any random sample of the training data set.

Point 11: The entire mathematical approach needs a revision according to the requirements of such scientific communication. I skipped the 4.3 paragraph until the paper will be revised in accordance with necessary rigor.

Response 11: We are so sorry to confuse you, and thank you for your proposal. On the basis of the previous sections, because the distribution interval of gradient is divided in this paper, we analyze the convergence in Section 4.3, and demonstrate whether the gradient descent in the division interval can achieve normal convergence without affecting the training. Therefore, this section is mainly the convergence proof of interval partition.

Point 12: In conclusion, the article discusses an important topic (distributed training), it provides one proposed approach (algorithm) which should be described clear, it provides a testing environment which is hard to reproduce and somehow not edifying for the intended  purpose of the algorithm (only two nodes which could not create bottleneck on communication, the synchronization is simple and it is intrinsic to the Tensorflow, thus, it is not quite what it would be needed).  The other algorithms against which the PGC was compared are not listed to show the “implementation” according to which the results where obtained. In order for the article to have value in this hot topic the experiments should be described as it can be reconstructed by other researcher the same as this article was able to implement the algorithms of other researchers.

Response 12: Thank you very much for your correction and suggestion. We have added the training parameters in Table 1 (Row 265) to facilitate the understanding and reproduction of the experiment. In this paper, our core idea is to propose a gradient compression algorithm, which acts on the optimization of transmission. Our goal is to expect that under the same communication conditions, this method can show better compression effect without affecting the accuracy of training. In addition, in the current open source TensorFlow framework, its test environment is not too difficult to implement. 

In this paper, we compare PGC with other algorithms in terms of convergence, compression ratio and training throughput. The details of the comparison can be seen in sections 5.1, 5.2 and 5.3. Meanwhile, we set up a reference method and a contrast method to carry out the experiment according to the Convention, so we only compare the three methods to verify our algorithm.

Reviewer 2 Report

The aim of this article is to develop a partition-based gradient compression algorithm for training and testing artificial neural networks used in IoT networks. This is a very interesting work on the complex problem of teaching IoT nodes. The work so far has focused on selected IoT nodes and methods of teaching them. However, the presented work tries to approach the issue of teaching and testing the Internet of Things nodes holistically. While many researchers also focus on the independent examination of individual nodes, the compression algorithm shows that one method can be adopted to train the entire IoT network. I have no comments on the content of the article, but it seems to me that the conclusions should be much broader, especially since those presented in the article show the direction in which research on the use of artificial intelligence algorithms in forecasting in IoT networks should develop. The article is fully suitable for publication. I am asking the authors to extend their applications.

Author Response

Point 1: The aim of this article is to develop a partition-based gradient compression algorithm for training and testing artificial neural networks used in IoT networks. This is a very interesting work on the complex problem of teaching IoT nodes. The work so far has focused on selected IoT nodes and methods of teaching them. However, the presented work tries to approach the issue of teaching and testing the Internet of Things nodes holistically. While many researchers also focus on the independent examination of individual nodes, the compression algorithm shows that one method can be adopted to train the entire IoT network. I have no comments on the content of the article, but it seems to me that the conclusions should be much broader, especially since those presented in the article show the direction in which research on the use of artificial intelligence algorithms in forecasting in IoT networks should develop. The article is fully suitable for publication. I am asking the authors to extend their applications. 

Response 1: Thank you very much for your correction and suggestion. We have added the relevant applications statement in Row 343.

Reviewer 3 Report

In this paper, the authors analyze and study the distribution characteristics of gradient parameters in neural network training for AIoT systems, and propose an adaptive compression strategy based on gradient partition.

My main concerns are as follows:

  1. AIOT term is used but the authors did not exploit it properly, it just remains in the title. I understood what the authors intended to show but they hardly touch IOT. So this reviewer strongly suggests incorporating the IOT concept.

2. Problem definition and challenges are not clear.

3. please explain your approach with theory and algorithms. Presently it just a large number of mathematical equations. That's not acceptable, for this journal and a large number of readers will not find this useful. Provide your algorithm and then establish the theory with mathematical eqns.

4. Same goes for experimental setup, where is IOT in the experimental parameters. What would be the changes in this result if we don't mention IOT? This reviewer found that IOT plays hardly any role in the result section, so authors should design experiments accordingly.

5. No state-of-the art comparisons w.r.t IOT, baseline fig 6 is too obvious.

6. Image quality of the paper is too poor.

Overall, this paper requires lots of modifications. If the authors cant address all the concerns in the scope of major revision, I would suggest submitting the paper ``as new''. But, extensive revision is required.

Author Response

Point 1: AIOT term is used but the authors did not exploit it properly, it just remains in the title. I understood what the authors intended to show but they hardly touch IOT. So this reviewer strongly suggests incorporating the IOT concept.

Response 1: Thank you very much for your correction and suggestion. We have added the relevant content of AIoT in the first paragraph of Introduction, and made Figure 1 to better explain the connection between this article and the IoT, hoping to get your approval.

Point 2: Problem definition and challenges are not clear.

Response 2: Thank you very much for your correction. Thank you very much for your correction. We have added relevant issues and challenges in the first paragraph of Introduction.

Point 3: please explain your approach with theory and algorithms. Presently it just a large number of mathematical equations. That's not acceptable, for this journal and a large number of readers will not find this useful. Provide your algorithm and then establish the theory with mathematical eqns.

Response 3: We are so sorry to confuse you, and thank you for your proposal. In Section 4.2 of this paper, we give the meaning of each parameter in the Algorithm 1, and give the complete Algorithm 1. The section 4.1 mainly deduces the information entropy formula of normal distribution, and section 4.3 proves whether the gradient descent can achieve normal convergence in the case of gradient partition. In addition, we reorganize the proof process in Section 4.3, hoping to make you and readers understand the relevant proof ideas more clearly.

Point 4: Same goes for experimental setup, where is IOT in the experimental parameters. What would be the changes in this result if we don't mention IOT? This reviewer found that IOT plays hardly any role in the result section, so authors should design experiments accordingly.

Response 4: Thank you very much for your correction. In this paper, we mainly discuss whether the gradient compression method can bring better transmission effect in the scene of distributed training. Therefore, the core of this paper is the optimization problem in transmission. As for the IoT is not involved in the experiment, it is only because the communication optimization between multi nodes and cloud in the IoT is modeled as the transmission optimization between worker nodes and parameter server in distributed training. As shown in Figure 1, although the real IoT environment is not simulated in the experiment, the idea to solve the problem is common. We hope that this algorithm can provide a good idea for solving the transmission optimization problem in AIoT. Therefore, the solution to the problem of transmission optimization is what we pay close attention to.

Point 5:  No state-of-the art comparisons w.r.t IOT, baseline fig 6 is too obvious.

Response 5: Thank you very much for your correction. Baseline is used as a reference method, and there is no compression strategy in itself, so the performance it shows is less than the other two compression methods.

Point 6:  Image quality of the paper is too poor.

Response 6: We are so sorry for this. We've improved the resolution of the images, hoping to meet your requirements.

Round 2

Reviewer 1 Report

Mistypes:

Row 30 – woeker

104-106 – the predicate is missing. I think it needs a coma instead of full stop, i.e. “… quantization schemes, they …”

196 – eq. (3) – the first log should also be ln

Again, I argue the reason of using chi square distribution… There is a well known result what is the entropy of a normal distribution and the proof doesn’t need the chi square, which, by the way has a positive support, not infinite. The last one is another confusion that throw some doubts regarding the mathematical fundaments. This is the reason why from the beginning I asked to omit some so called profs from the article.

The 4.1 through 4.3 sections

I will try to resume:

In eq (2) it is supposing (without a clear statement) that x is a probabilistic variable which semes to be of real value to be introduced in that particular equation.

 In algorithm 1 we have G l as the gradient, let say we chose layer 1, this mean that G1 is a vector. Then we have line (224) where x is the gradient, this mean x is a vector, not the x previously mentioned . Then we have eq (16) where we have the absolute value of x, which could mean the module of the vector. Though, sign(x)*alpha should be defined if x is a vector. Then we have line(234) where we have “any random sample of training data set is represented as phi(x,y).” Who would be the x and y in this new context, because now x is not the gradient?

I remain to my opinion that the math doesn’t fulfil the necessary rigorousness.

Author Response

Point 1: Row 30 – woeker

Response 1: We are so sorry for this and thank you very much for your correction. We have corrected this error.

Point 2:104-106 – the predicate is missing. I think it needs a coma instead of full stop, i.e. “… quantization schemes, they …”

Response 2: Thank you very much for your correction. We have made changes according to your suggestions and hope to meet your requirements.

Point 3:196 – eq. (3) – the first log should also be ln

Response 3: Thank you very much for your suggestion and we have corrected it in eq. (3).

Point 4:Again, I argue the reason of using chi square distribution… There is a well known result what is the entropy of a normal distribution and the proof doesn’t need the chi square, which, by the way has a positive support, not infinite. The last one is another confusion that throw some doubts regarding the mathematical fundaments. This is the reason why from the beginning I asked to omit some so called profs from the article.

Response 4: We are so sorry to confuse you and thank you for your correction. We have deleted some unnecessary proofs. Indeed, the entropy of the normal distribution should be well known. It's a bit redundant here. Thank you again for your correction and hope it meets your requirements.

Point 5: The 4.1 through 4.3 sections, I will try to resume: In eq (2) it is supposing (without a clear statement) that x is a probabilistic variable which semes to be of real value to be introduced in that particular equation.

In algorithm 1 we have G l as the gradient, let say we chose layer 1, this mean that G1 is a vector. Then we have line (224) where x is the gradient, this mean x is a vector, not the x previously mentioned . Then we have eq (16) where we have the absolute value of x, which could mean the module of the vector. Though, sign(x)*alpha should be defined if x is a vector. Then we have line(234) where we have “any random sample of training data set is represented as phi(x,y).” Who would be the x and y in this new context, because now x is not the gradient?

Response 5: We are so sorry to confuse you and thank you for your questions. In eq (2), x is only a representation of function and has no substantial meaning. In line 224, we use x to represent the gradient, which brings you and readers great confusion. We have changed the representation of gradients to avoid confusion and make it easier for readers to understand our paper (the latest revision is in line 218).

Phi(x,y) in line 234 is only a representation of any one random sample in the dataset, which has nothing to do with the gradient value in this paper. We are so sorry for confusion caused by the mistake in writing, the correct expression should be j : (x, y), where (x, y) is the form of any one random sample j.

In the neural network training, the data set is generally composed of training set and verification set, it is usually expressed in the form of (x, y). Where x represents a training instance in the training set, and y represents the correct result of x known in the verification set. We input the data in the training set into a network model, and compare the output result with the correct value y known, so as to judge the quality of the training network model. Therefore, j : (x, y) in line 234 is only the form of any one random sample in the dataset, which has nothing to do with the gradient value in this paper.

In order to avoid the confusion with x in the previous example, we only express any one random sample as just j (in line 228), without adding its representation (x, y). Hope to solve your question, thank you again for your correction.

Reviewer 3 Report

I find that revisions have addressed the reviewers' concerns.

Author Response

Thank you very much for your comments, we hope to meet your requirements.